# Design of hidden thermodynamic driving for non-equilibrium systems via mismatch elimination during DNA strand displacement

Natalie E. C. Haley [1], Thomas E. Ouldridge [2]✉, Ismael Mullor Ruiz[2], Alessandro Geraldini[3], Ard A. Louis[3], Jonathan Bath[1] & Andrew J. Turberfield [1]✉

Recent years have seen great advances in the development of synthetic self-assembling molecular systems. Designing out-of-equilibrium architectures, however, requires a more subtle control over the thermodynamics and kinetics of reactions. We propose a mechanism for enhancing the thermodynamic drive of DNA strand-displacement reactions whilst barely perturbing forward reaction rates: the introduction of mismatches within the initial duplex. Through a combination of experiment and simulation, we demonstrate that displacement rates are strongly sensitive to mismatch location and can be tuned by rational design. By placing mismatches away from duplex ends, the thermodynamic drive for a strand-displacement reaction can be varied without significantly affecting the forward reaction rate. This hidden thermodynamic driving motif is ideal for the engineering of non-equilibrium systems that rely on catalytic control and must be robust to leak reactions.

[1] Clarendon Laboratory, Department of Physics, University of Oxford, Parks Road, Oxford OX1 3PU, UK. [2] Imperial College Centre for Synthetic Biology and Department of Bioengineering, Prince Consort Road, Imperial College London, London SW7 2AZ, UK. [3] Rudolf Peierls Centre for Theoretical Physics, Department of Physics, University of Oxford, Keble Road, Oxford OX1 3NP, UK. ✉email: t.ouldridge@imperial.ac.uk; andrew.turberfield@physics.ox.ac.uk

One of the signature features of living systems is that they operate continuously, expending free energy, rather than relaxing to equilibrium. Key molecular components are not consumed by these reactions but recovered—they act as catalysts[1]. Examples include metabolic enzymes, signal-processing kinases, molecular motors, polymerases and even nucleic acids undergoing replication, transcription and translation[2].

Designing synthetic analogues of such complex molecular systems is a key goal of nanotechnology. This task is complicated by the requirements that reactions must be thermodynamically driven in the desired directions and that stable equilibrium states that lock up the key catalytic components, and accidental leak reactions, must be avoided. Overall reaction thermodynamics and kinetics must therefore be carefully tuned.

Due to its predictable base-pairing interactions, DNA has proved to be a remarkably successful material for the construction of static nanoscale structures[3–7]. Toehold-mediated strand displacement (TMSD)[8] and toehold exchange[9], processes in which an invading strand replaces another strand in a DNA duplex by competing for base pairs, have allowed for the development of dynamic DNA-based systems. Many examples operate in a single-shot fashion in response to an input[10,11]. Continuously operating dynamic systems, including reaction network architectures[11–13] and synthetic molecular machinery[14–16], have also been developed, but the state-of-the-art is far from matching the power and flexibility of natural analogues.

As we outline in this article, one of the main challenges with implementing catalysis in DNA-based systems is that a thermodynamic drive towards the product, which is necessary to ensure high yields in reaction steps, is typically achieved through the formation of additional base pairs in the product. However, these added base pairs tend to accelerate leak reactions, in which substrates are converted into products in the absence of the catalyst, compromising our ability to exert catalytic control over the reaction rate. An ideal solution to the problem that leak reaction rates and free energy changes are coupled would be to provide a hidden thermodynamic advantage that makes conversion of substrates into products substantially more favourable, without significantly affecting the leak reaction rate.

We propose the use of internal base-pairing mismatches within DNA duplex substrates to provide hidden thermodynamic driving. Through a combination of experiment and simulation, we demonstrate that, although an internal mismatch substantially destabilises a duplex, its influence on the strand-displacement rate is highly position dependent: the mismatch location can be chosen rationally to provide a hidden thermodynamic drive. We find that reaction rates are highest when the mismatch is eliminated early, but not immediately, during strand exchange: introduction of an initial mismatch can increase the reaction rate by approximately two orders of magnitude, whereas later mismatches have a very small effect. We use numerical simulations to explain this behaviour and confirm that late mismatches constitute a hidden thermodynamic drive. We demonstrate that mismatch elimination indeed allows for both tight catalytic control and strong thermodynamic driving in a simple catalytic system, and exploit the mismatch location dependence of strand-displacement rates in a DNA pulse generator.

## Results

### The need for hidden thermodynamic driving in DNA networks.
TMSD[8] and toehold exchange[9], the key reactions underlying much of DNA nanotechnology, are illustrated in Fig. 1a, b. A double toehold exchange process provides a simple mechanism for DNA-mediated catalysis (Fig. 1b): the invader $A$ catalyses the replacement of $B$ by $C$ in the duplex with strand $D$. Figure 1d

shows the dramatic increase in the rate of this reaction that results from addition of a small concentration of catalyst $A$, compared to the same reaction with no catalyst. We can use such catalytic mechanisms to control the rates of hybridisation reactions[14].

This simple mechanism also illustrates a general limitation to the use of catalytic triggers to control reactions: $A$ also accelerates the reverse reaction. The equilibrium yield, which is determined by the balance between forward and reverse reaction rates, is unaffected by catalysis. In the toehold-exchange reaction shown in Fig. 1b, the standard free energy of product $CD$ is close to that of the initial duplex $BD$, which contains the same number of base pairs: this fact limits the yield of the reaction in isolation. If the reaction is to be driven close to completion (100% conversion of substrate into product), a thermodynamic driving force must be provided, either by supplying a large excess of the invading strand (which would increase the "leak" reaction rate in the absence of the catalyst), or by sequestering the output in a downstream complex. The latter strategy is applied in the experiment shown in Fig. 1d, in which a reporter system (Fig. 1c) reacts with free $B$ strands as they are produced: the reaction therefore slowly approaches completion. The absence of an intrinsic thermodynamic drive within the catalytic motif itself can present a significant challenge, when combining these motifs within large networks: analogues in living systems can couple catalysis directly to the turnover of high-free-energy molecules, such as ATP.

We could increase the yield of the system in Fig. 1b by allowing $C$ to bind to more bases of $D$ than $B$ does, making $\Delta G_{BD \to CD}$ more negative. However, if these bases provide a toehold for $C$ to initiate binding to the $BD$ duplex, they can enhance the ability of $C$ to displace $B$ without the intervention of $A$: this leak reaction compromises our ability to exert catalytic control over the reaction rate. We confirm this hypothesis in Fig. 1d, in which an altered strand $C_2$ that can form two extra base pairs with $D$ is shown to displace $B$ rapidly even in the absence of the catalyst $A$.

We propose that adding mismatched base pairs to the $BD$ duplex will provide a hidden thermodynamic drive, making $\Delta G_{BD \to CD}$ more negative while having only a minimal effect on leak reaction kinetics. To explore this possibility, we first investigate how mismatch placement within an incumbent duplex influences reaction rate.

### The dependence of displacement rate on mismatch position.
Experiments to measure mismatch-elimination TMSD reaction kinetics are shown in Fig. 2. Substrate duplexes $OT$ contain a mismatch that is eliminated by the invading strand $I$ (Fig. 2). The final duplex $IT$ thus has perfect complementarity. The toehold on the substrate duplex was chosen to be four nucleotides (nt) long and the displacement domain of $OT$ was 20 base pairs (bp) in all cases. Single cytosine–cytosine (C–C) mismatches were inserted in the displacement domain at positions between 2 and 17 bases from the toehold. To ensure the maximum degree of comparability between these systems, designs with mismatches at positions $2 \to 3$ were obtained by translating the mismatch sequence along the $OT$ duplex, and mismatches at positions 15 and 17 were obtained by single-point mutations to the target strand.

A fluorescent reporter reaction was used to measure displacement rates[9,17–19]. Reporter fluorescence is activated by single-stranded $O$ displaced from $OT$. Strand $O$ possesses a 5′ single-stranded domain that forms the displacement domain of the reporter reaction, but the toehold-binding domain that initiates this reaction is sequestered in the $OT$ duplex until $O$ is released. Liberated $O$ displaces a fluorophore-labelled strand from the reporter duplex, in which the fluorophore is quenched (Fig. 2). The reporter reaction is thus also based on the TMSD mechanism,

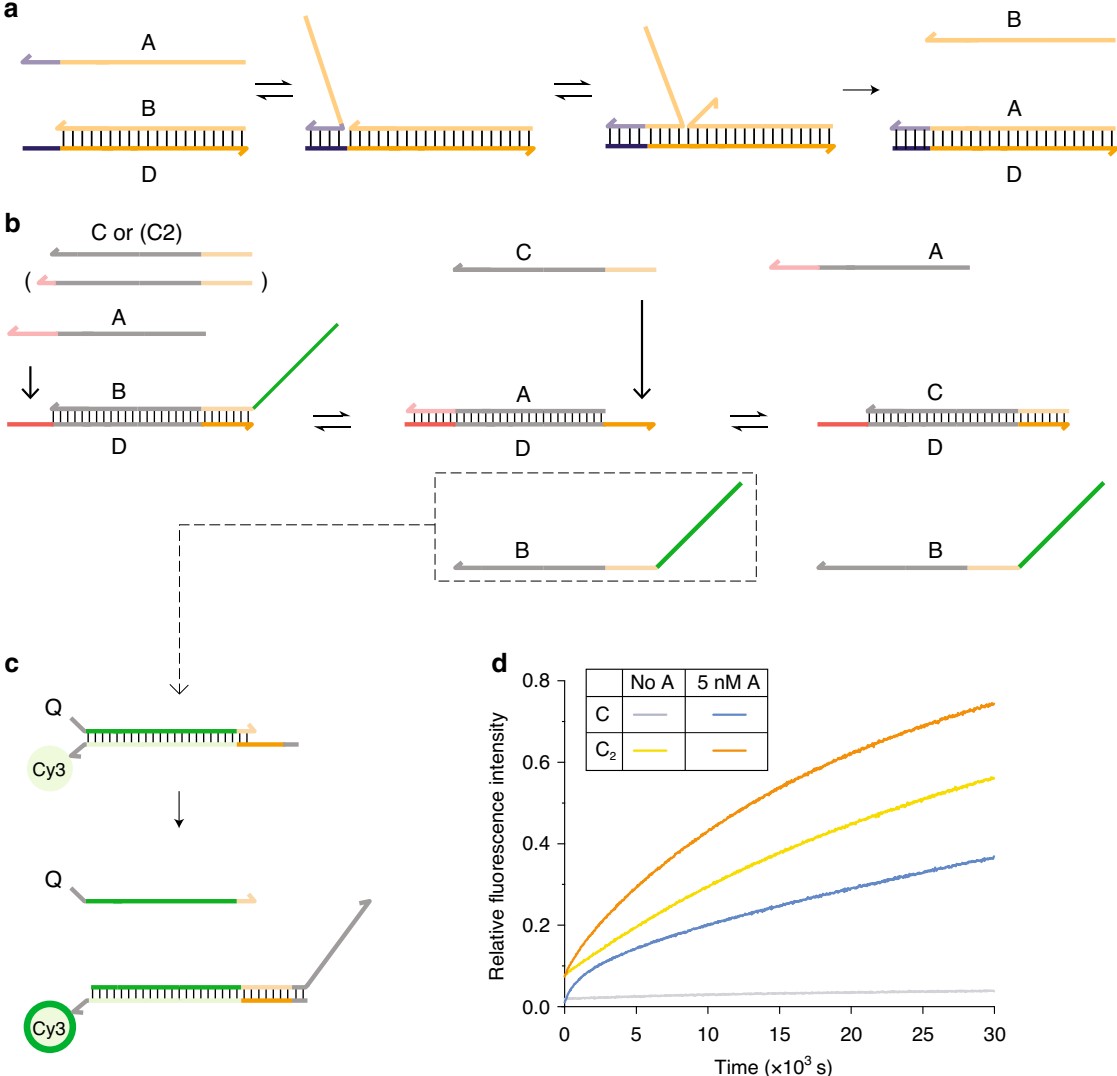

**Fig. 1 Strand displacement and catalysis in DNA nanotechnology. a** Toehold-mediated strand displacement. The reactant duplex *BD* can be attacked by an invading strand *A* that is complementary to the whole of *D*, including an initially single-stranded toehold. **b** A simple catalytic motif engineered through toehold exchange. Displacement of *B* by *A* (which are complementary to different but overlapping domains of *D*) exposes a second toehold that allows the subsequent displacement of *A* by *C*. *A* can thus act as a catalyst for the interconversion of *BD* and *CD*, providing control over the reaction. **c** Addition of a fluorescent reporter for single-stranded *B* allows the progress of the reaction to be monitored. **d** Normalised fluorescence data showing the effect of adding catalyst *A* on the catalytic scheme in **b** with 22 bp duplexes and 6 nt toeholds. A total of 40 nM of *BD* is mixed with 200 nM of *C* and 250 nM of reporter complex, both with and without triggering by addition of 5 nM of *A*. Output in the absence of *A* (the leak reaction) is low, but the reaction yield after 500 minutes is limited, a problem exacerbated by the similar stabilities of reactant and product. A longer strand $C_2$ that can form two extra base pairs with *D* (in the toehold domain for binding of *A*) increases yield over 500 minutes but at the cost of a loss of catalytic control: the leak reaction in the absence of the catalyst is now extremely strong. Each curve represents a single experiment; a repeat is shown in Supplementary Fig. 7. A full list of sequences is provided in Supplementary Table 6.

but with a longer toehold of 7 nt to ensure a significantly greater reaction rate. The reporter thus provides a signal proportional to the cumulative output of *O* from *OT* at any given time (see Supplementary Note 1.1).

Fitted rate constants $k$ for the mismatch-elimination reaction varied between $k \sim 1.8 \times 10^5 \, M^{-1}s^{-1}$ for mismatch position 3 and $k \sim 1.7 \times 10^3 \, M^{-1}s^{-1}$ for position 15. Adjustment of the mismatch elimination position can thus change the displacement rate by approximately two orders of magnitude. The rate constant for an equivalent system with no mismatches is $k_0 = 2.6 \times 10^3 \, M^{-1}s^{-1}$. Relative rates are plotted in Fig. 2b. The reaction rate constant rises as the mismatch is moved away from the toehold up to position 3; it then falls approximately exponentially back to the mismatch-free rate for positions $\gtrsim 13$. This non-monotonicity

differs from the monotonic behaviour observed in a previous study of kinetic control through mismatch creation[19].

**Insights from coarse-grained simulations of DNA.** The results in Fig. 2 suggest that mismatches far from any duplex ends could be used to provide hidden thermodynamic driving: the effect of the introduction of the base-pair mismatch on the reaction rate is limited despite a large predicted change in the free energy of the reaction of up to 6 kcal mol$^{-1}$ (ref. [20]), the contribution of three to four extra base pairs. To investigate this possibility, and to aid the rational design of such systems, we simulated mismatch-elimination displacement with a coarse-grained model, oxDNA (refs. [21,22]). oxDNA describes DNA as a string of rigid nucleotides with pairwise interactions that represent excluded volume,

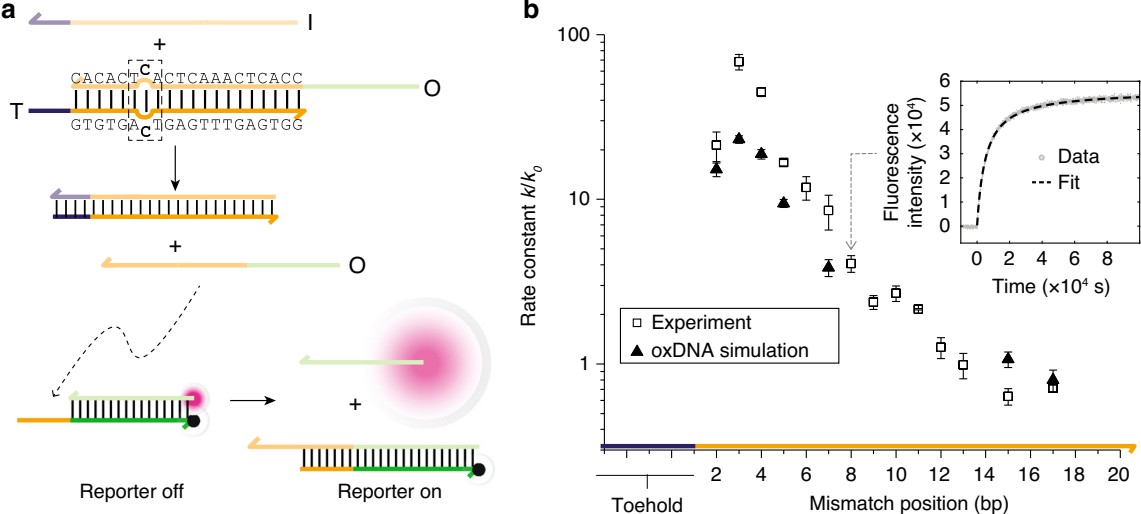

**Fig. 2 The position of an eliminated mismatch affects strand-displacement rate. a** Mismatch-elimination TMSD: an invader *I* displaces an output strand *O* from an *OT* duplex containing a mismatch. *O* subsequently participates in a secondary displacement reaction that liberates a fluorophore-bearing strand from a quenching duplex. In this example, complex *OT* has a cytosine–cytosine mismatch at position 7, counting from the beginning of the displacement domain. The three-base-pair mismatch motif (dashed rectangle) is the same for all mismatch positions. **b** Experimentally obtained TMSD rate constant as a function of mismatch position relative to $k_0 = 2.6 \times 10^3 \, M^{-1}s^{-1}$ for a mismatch-free system (squares). Fluorescence data and fit for position 7 is shown inset. Triangles show corresponding relative rate constants obtained from oxDNA simulations, showing a similar non-monotonic relationship between reaction rate and mismatch location. Error bars show standard error on the mean ($2 \leq n \leq 6$ for experiment, $n = 6$ for simulation); there is also an error that systematically effects all relative rates due to the error in determination of no-mismatch rate constant $k_0$ of ±15% for experimental points ($n = 7$) and ±11% for simulated points ($n = 6$).

backbone connectivity, base-pairing and stacking interactions. It is simple enough that simulations of complex processes, such as strand displacement, are feasible while retaining enough of the underlying physics to capture the thermodynamic, mechanical and structural changes associated with the formation of duplex DNA from single strands. oxDNA has previously been applied to a number of systems in which displacement is important[19,23–25]. Here, we use the sequence-dependent parameterisation of Sulc et. al.[22].

Using oxDNA with forward flux sampling, as outlined in the "oxDNA methods" section, and Supplementary Notes 4.1 and 4.2, we obtained predictions for displacement rate constants for mismatch positions 2, 3, 4, 5, 7, 15 and 17 relative to the mismatch-free case (note that estimates of relative rates obtained from coarse-grained models are more accurate than absolute rates[23,26]). Results are plotted in Fig. 2b, and a snapshot of the simulated displacement process is shown in Fig. 3a. oxDNA reproduces the initial rise in rate up to mismatch position 3, followed by the decay back to the mismatch-free rate for late mismatch positions. The maximal acceleration, by a factor of 23, is comparable to, but smaller than, that observed in experiment. Much of this discrepancy may be due to known features of the oxDNA model: a slight enhancement of the tendency of short toeholds to mediate successful displacement[23,25] (limiting the potential speed up due to mismatch elimination), and a slight underestimation of the destabilisation of DNA due to mismatches[21].

More importantly, oxDNA elucidates the biophysical mechanism underlying the observed behaviour. In the second-order limit, relevant for weak, reversible toehold binding, the overall rate constant can be rewritten as

$$k = k_{\text{toehold}} p(\text{disp}|\text{toehold}), \quad (1)$$

where $k_{\text{toehold}}$ is the rate of toehold binding and $p(\text{disp}|\text{toehold})$ is the probability of successful displacement (rather than abortive detachment) once the toehold duplex has formed. Given that the toehold is the same for all systems studied, $k_{\text{toehold}}$ should be similar for all systems (as found in oxDNA simulations: see Supplementary Note 4.2). The key quantity is therefore $p(\text{disp}|\text{toehold})$. Results for the first-order limit, which may be relevant at extremely high effective concentrations of reactants, such as for surface-bound schemes[27], are presented in Supplementary Note 1.2.

We can split $p(\text{disp}|\text{toehold})$ further into

$$p(\text{disp}|\text{toehold}) = p(\text{disp}|\text{bp } x \text{ reached}) p(\text{bp } x \text{ reached}|\text{toehold}), \quad (2)$$

where $x$ is any base-pair position in the displacement domain. The probabilities $p(\text{disp}|\text{bp } x \text{ reached})$ and $p(\text{bp } x \text{ reached}|\text{toehold})$ are plotted in Fig. 3b for the mismatch-free system. Also shown are the probabilities for reaching position $x$, and the probability of displacement given that $x$ has been reached, for a system with a mismatch at $x$. In the first case, the product is independent of the choice of reference base-pair $x$; in the second, it is strongly dependent on the position $x$ of the mismatch. We see that, depending on the value of $x$, a mismatch at $x$ can enhance both the probability that base-pair $x$ is reached and the probability of successful displacement given that $x$ has been reached. Mismatches weaken the duplex $OT$, accelerating branch migration to the position of the mismatch, and discourage reverse branch migration once the elimination is made. Both effects accelearate strand displacement. Both are suggested by the free-energy profile of displacement, which quantifies the stability of intermediate states as a function of reaction progression, plotted in Fig. 4. Lower free energies indicate thermodynamically favoured states: it can be seen from Fig. 4 that the forward progress of displacement is encouraged by a large drop in free energy, and that this drop begins slightly before the mismatch is actually enclosed.

The magnitudes of these effects depend strongly upon $x$. For relatively late mismatches ($x \gtrsim 7$), the existence of the mismatch has almost no effect on the probability that this position is reached by the invader. The destabilising effect of the mismatch is quite local, affecting only nearby base pairs (Fig. 4): if the

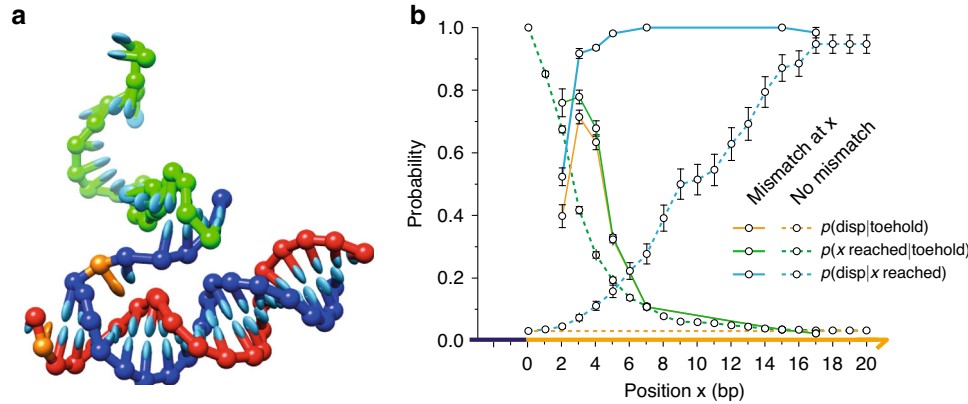

**Fig. 3 oxDNA simulation of mismatch-elimination TMSD.** **a** Snapshot from an oxDNA simulation showing the invader (green) binding to the toehold of the target (blue); the output (red) is bound to the displacement domain. The mismatched base pair, highlighted in orange, is at position 2; fraying of the *OT* duplex is visible. **b** Analysis of strand-displacement probabilities using oxDNA. The probability that the branch point reaches position *x* (*p*(*x* reached|toehold)) and the probability of successful displacement given that *x* is reached (*p*(disp|*x* reached)) are plotted for a perfectly matched duplex (dashed green and blue lines, respectively) and a duplex with a mismatch at *x* (solid green and blue lines, respectively). Also shown is the overall displacement probability *p*(disp|toehold) for the perfectly matched case and *p*(disp|toehold) as a function of mismatch location (dashed and solid orange lines, respectively). Error bars show standard error on the mean for *n* = 6. The presence of a mismatch at position *x* enhances both the probability of reaching *x* and of successful displacement given that *x* is reached.

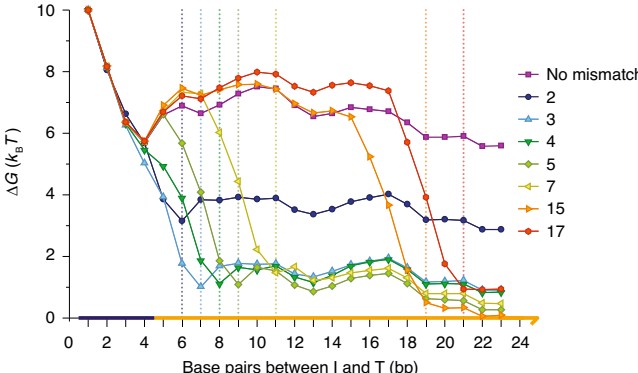

**Fig. 4 Free-energy profile of strand displacement.** Simulated configurations are assigned to macrostates based on the number of base pairs between *I* and *T*. Mismatch elimination is associated with a large drop in free energy that begins slightly before mismatch enclosure (dotted lines indicate the expected number of base pairs at the point of mismatch elimination). A mismatch at position 2 provides a smaller overall thermodynamic bias for displacement than mismatches at other positions. Estimated errors for individual points within a single simulation are highly correlated with other points. The overall free energy change between 1 and 23 base pairs therefore gives the best characterisation of the sampling error: we find a typical standard error of ~±0.2$k_B T$ from *n* = 6 independent simulations (see Supplementary Table 23 for details).

mismatch is sufficiently far from the toehold, altering the last few steps prior to the mismatch has almost no effect on the probability of reaching it. Additionally, once mismatches at large values of *x* are enclosed, the probability of eventual displacement is essentially unity. However, the probability of displacement having reached larger values of *x* deep within the displacement domain is high even without the mismatch, so the overall increase in *p*(disp|toehold) due to the mismatch is quite small. The displacement rate therefore converges with the mismatch-free case for large *x*.

For early mismatches (small *x*), the presence of the defect significantly increases both *p*(disp|bp *x* reached) and *p*(bp *x* reached|toehold), suggesting that low values of *x* are optimal for

accelerating displacement. However, *p*(disp|bp 2 reached) is much smaller than *p*(disp|bp 3 reached), leading to a substantial difference in rates and the non-monotonic dependence of displacement speed on mismatch position. The reason for this behaviour is that a mismatch at *x* = 2 is significantly less destabilising than a mismatch at *x* = 3 (refs. [21,28]). Instead of fully enclosing the mismatch at position 2, the terminal base pair in the *OT* duplex can fray (Fig. 4), exchanging the high cost of incorporating a mismatch into the duplex for the lower cost of disrupting the terminal base pair. As a result, eliminating the mismatch provides a smaller thermodynamic advantage for *IT* over *OT* than eliminating a mismatch deeper inside the duplex, as can be seen by comparing the position 2 curve to the others in the free-energy profiles of Fig. 4. A mismatch eliminated at position 2 thus provides a smaller barrier to reversal of branch migration than at later points, hence *p*(disp| 2 reached) is relatively small.

The free-energy profiles in Fig. 4, combined with the displacement rate data in Fig. 2b, indicate that mismatch elimination can provide hidden thermodynamic driving when positioned appropriately. All eliminated mismatches make a large, negative contribution to *ΔG* of reaction, but mismatches encountered later during branch migration have a very limited effect on reaction rates.

**Generalisation to alternative toehold lengths.** We have studied relatively weak four-nucleotide toeholds, which also serve as a proxy for a partially matched longer toeholds, demonstrating our ability to enhance *ΔG* in such cases without substantially increasing leak reaction rates. However, the understanding gained from oxDNA allows us to predict three broad regimes of response to mismatch elimination, depending on the toehold length and concentrations of invading strands. These regimes are: (i) the second-order limit in which the three-stranded complex *IOT* complex is short-lived and the probability of resolving into *IT* + *O*, *p*(disp|toehold), is intrinsically low; (ii) the second-order limit in which the *IOT* complex is short-lived and *p*(disp|toehold) is intrinsically high; and (iii) the first-order, high-concentration limit in which concentrations are high enough that the toehold-binding rate exceeds the rate at which *IOT* resolves into either *I* + *OT* or *IT* + *O*.

The systems studied above fall into regime (i), with short toeholds and low concentrations of strands. In this case, as

outlined by Srinivas et al.[23], the reaction rate constant is given by $k_{\text{toehold}}p(\text{disp}|\text{toehold})$, with $k_{\text{toehold}}$ the rate constant for binding to the toehold. For short toeholds, mismatch-free systems have a low $p(\text{disp}|\text{toehold})$, but we have shown that the strand-displacement probability can, in principle, be substantially enhanced by mismatches that are eliminated during the displacement process if the mismatch is encountered early during displacement. Length-zero toeholds are a particularly relevant limit in this class, as this motif occurs frequently in the design of catalytically controlled, kinetically frustrated systems[12,13,29]. In this case, both ends of the strand are equivalent, so mismatches at either end are effectively early and will significantly accelerate strand displacement. We have shown that centrally located mismatches in long displacement domains provide only weak enhancement of both conditional probabilities $p$ ($\text{disp}|\text{bp } x \text{ reached}$) and $p(\text{bp } x \text{ reached}|\text{toehold})$, leading to an enhancement of displacement rates by only a factor of ~2 despite their large contribution to $\Delta G$: they therefore provide effective hidden thermodynamic driving.

Regime (ii) corresponds to the use of relatively long toeholds and low strand concentrations. In this case, $p(\text{disp}|\text{toehold})$ is saturated (is ~1) since dissociation of the toehold is extremely rare. Hence, we expect no acceleration of the reaction by mismatches. This prediction is consistent with the results of Broadwater et al., who observed no kinetic consequences of mismatch elimination when using toeholds of length 10 (ref. [30]).

In regime (iii), $p(\text{disp}|\text{toehold})$ becomes irrelevant since a failed invader is rapidly replaced with another. Instead, the reaction rate is dominated by the time required to reach $IT + O$ from the toehold-bound state. Based on additional results from oxDNA (Supplementary Note 1.2, Supplementary Fig. 6, Supplementary Tables 11 and 12), we predict that mismatch elimination can reduce this time, by up to a factor of 3, when mismatches are encountered early, but not too early, during the displacement process. In this context, mismatch elimination might be used to increase the fundamental speed limit of displacement reactions occuring, for example, at high effective concentrations between surface-localised reactants[27].

**Hidden thermodynamic driving in a catalytic motif**. We now return to the catalytic motif of Fig. 1b, and demonstrate that an initial mismatch in the $BD$ complex that is eliminated in the $CD$ complex can substantially enhance reaction yield without severely compromising catalytic control. To test the generality of the mismatch-elimination TMSD motif, we consider two initial mismatches types, C–A and T–T, which become C–G and T–A, respectively, after mismatch elimination. Both are distinct from the C–C mismatches used in the experiments presented in Fig. 2b. These mismatches are created by changing the sequences of $A$ and $B$ to create a mismatch close to the centre of the initial $BD$ complex (Fig. 5a); we label the changed strands $A_{CA}$, $A_{TT}$, $B_{CA}$ and $B_{TT}$, where the subscript corresponds to the mismatched base pair in the initial duplex $BD$ that persists in intermediate $AD$ and is eliminated in product $CD$. Full sequences are provided in Supplementary Table 6.

We repeat the protocol of Fig. 1c: 40 nM of $B_{XX}D$ is mixed with 5 nM of $A_{XX}$ and 200 nM of $C$, where XX is either CA or TT; the leak reaction is assessed by omitting $A_{XX}$. Flouresence of a reporter complex that binds to all variants of $B$ is used to report on the reaction. Fig. 5 shows strand-displacement kinetics for catalysed (Fig. 5b) and uncatalysed (Fig. 5c) strand displacement with mismatch elimination, alongside comparable data from Fig. 1c for mismatch-free systems with and without an addition 2-base toehold for $C$. Both mismatch elimination systems show substantially stronger rate enhancement in response to the

introduction of the catalyst than does the default system with perfectly matched sequences. We see that hybridisation catalysis is effective in this system even though the intermediate formed between strand $D$ and catalyst $A$ retains the initial mismatch, at least in the dilute limit with centrally placed mismatches. In fact, mismatch-elimination systems are substantially faster overall, faster even than mismatch-free system with the longer strand $C_2$ that forms more base pairs with $D$ (cf. Fig. 1). Crucially, however, the leak reaction in the absence of the catalyst remains weak, much weaker than when the longer strand $C_2$ is used (Fig. 5c), confirming that the thermodynamic drive is indeed successfully hidden. A second replica of the same experiments, with similar results, is reported in Supplementary Fig. 7.

In these experiments, the reporter complex acts as a sink for liberated $B$ (or $B_{TT}$ or $B_{CA}$) strands. The result is that all reactions should eventualy reach full substitution of $B/B_{TT}/B_{CA}$ by $C$, even when the initial catytic exchange is not thermodynamically driven towards completion. In Supplementary Note 1.3.2, Supplementary Fig. 8, we report data on catalyst-triggered systems that are left to run for 3 or 6 h before the reporter is added. Free $B$ strands that have already been liberated by catalytic exchange are consumed rapidly by the reporter, giving a fast initial increase in fluorescence that provides a measure of the progress of the reaction to that time. Subsequent reactions produce more free $B$ strands that are consumed by the reporter on a much longer timescale. For the mismatch-free system with no designed thermodynamic drive, reactions reach only 20–30% completion after both 3 and 6 h, consistent with the expected lack of a bias towards the product. By contrast, for the system driven by the C–A mismatch, the substrate is essentially fully reacted after 3 h, showing the efficacy of the C–A mismatch as a thermodynamic driver. The systems with T–T mismatch elimination and formation of two extra base pairs show intermediate behaviour.

**Pulse generation via differential acceleration of reactions**. Having successfully demonstrated hidden thermodynamic driving, we now explore the alternative possibility of designing systems in which the increase in reaction rate caused by early mismatches is also advantageous. We construct a simple pulse-generating device, in which two invaders compete to bind to a target using toeholds at opposite ends of the duplex. We label the invading strand that binds to the toehold at the 5′ end of the substrate duplex $I_5$, and the strand that binds to the 3′ end $I_3$ (as shown in Fig. 6a). Each invading strand bears a quencher at its toehold end and a fluorophore at the other (fluorophore–quencher pairs are Cy3/Iowa-BlackFQ and Cy5/IowaBlackRQ for $I_3$ and $I_5$, respectively). Given the particular flexibility of short ssDNA oligonucleotides[21], and their tendency to form transient hairpin structures, the fluorophore and quencher are, on average, in close proximity in the single-stranded state, leading to effective quenching. Binding to the target enhances the fluorescence from the successful invader as the formation of the target duplex separates the fluorophore from the quencher.

The two toeholds are designed to be as similar as possible, but asymmetry is built into the system by incorporating a mismatch in the initial duplex at a position 18 base pairs from the 5′ toehold and 3 base pairs from the 3′ toehold. The mismatch is thus encountered early by the 3′ invader and late by the 5′ invader. In these experiments, we use a fourth type of mismatch (C–T is corrected to A–T), demonstrating the robustness of the hidden thermodynamic driving motif.

We expect that the mismatch will provide an additional thermodynamic advantage for both invaders, but that it will substantially speed up invasion by the 3′ invader only. Our hypothesis is that the slowdown of late mismatch elimination

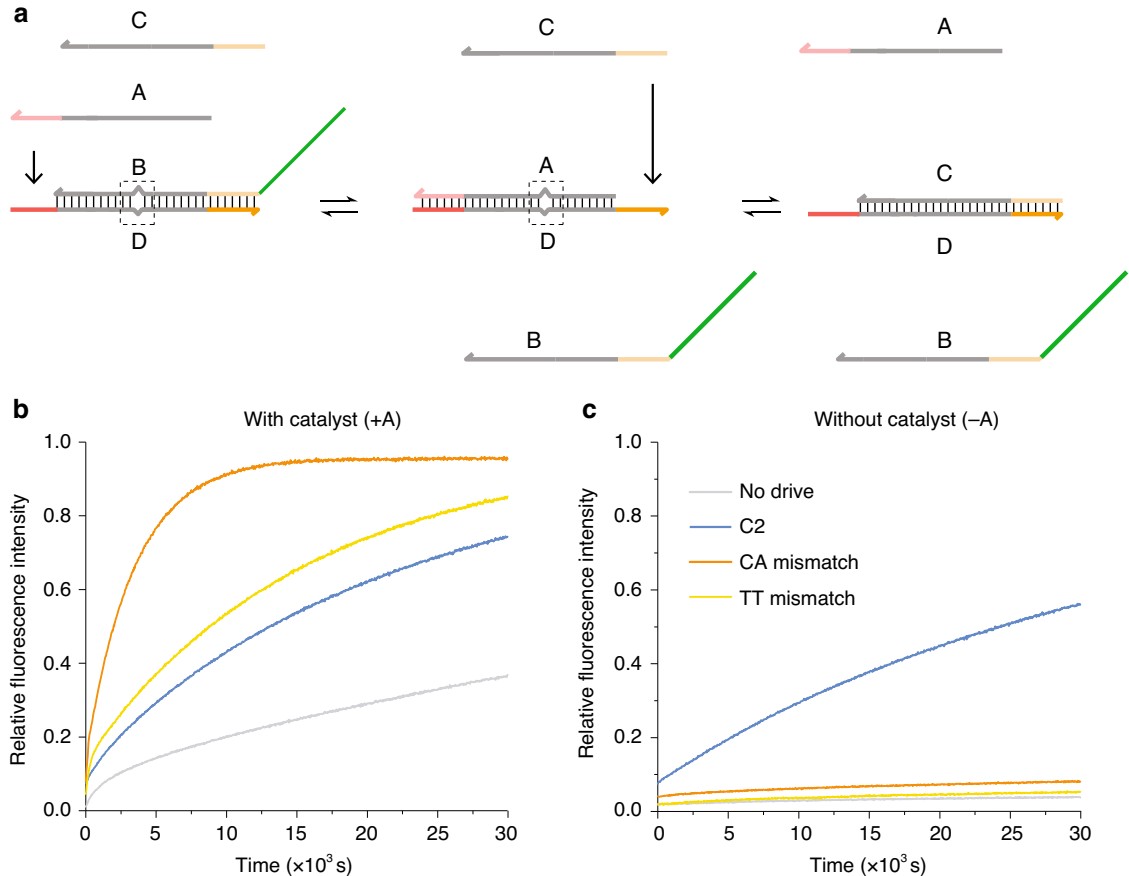

**Fig. 5 Mismatch elimination can provide hidden thermodynamic driving. a** Schematic of mismatch elimination during toehold exchange; a mismatch in the *BD* duplex is retained in the *AD* duplex, but eliminated in the final *CD* duplex. **b** Normalized fluorescence of a reporter for single-stranded *B* after 40 nM of *BD* is mixed with 200 nM of *C* and triggered by addition of 5 nM of catalyst strand *A*. Reactions presented are a no-mismatch system in which *BD* and *CD* have the same number of base pairs (no drive), a variant with a strand $C_2$ that can form two additional base pairs with *D* (C2), and two systems in which modified *A* and *B* strands form C–A and T–T mismatches with *D* that are eliminated in the *CD* duplex. These mismatches are positioned 12 and 11 base pairs, respectively, from the end of the *AD* duplexes. The systems in which mismatch elimination is used to provide a thermodynamic drive show strongly enhanced reaction rate and yield. **c** Monitoring of leak reactions in systems that are equivalent to **b**, but without the addition of the catalyst strands. The mismatch elimination systems show leak reaction rates that are comparable to the no drive system and much lower than the $C_2$ variant. Each curve shows a single experiment; a second replica is reported in Supplementary Fig. 7.

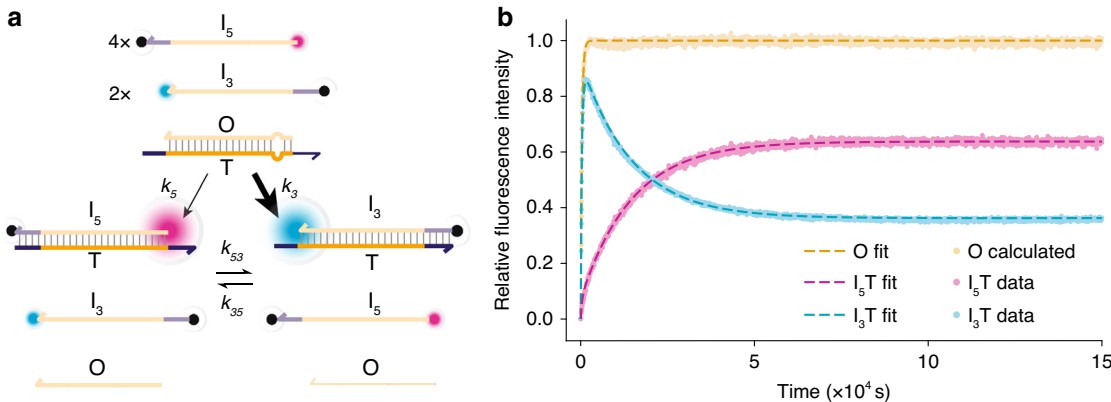

**Fig. 6 Pulse generation with two-toehold competitive displacement reactions. a** Schematic for two-toehold competitive displacement reaction with an asymmetrically placed mismatch in the substrate duplex *OT*, 18 bases from the 5′ toehold and 3 bases from the 3′ toehold. The 3′ invading strand $I_3$ should be kinetically favoured. 5′ invading strand $I_5$ is thermodynamically favoured as it is present in greater excess than $I_3$, and both eliminate the initially present mismatch. **b** T–C→T–A mismatch elimination at position 18 from the 5′ toehold of *OT*. Both processed data and ODE fits are shown for a single experiment (for fitted parameters see Supplementary Table 1). Initial relative concentrations are $[I_5]:[I_3]:[OT] = 4:2:1$. Note that the *O* curve is inferred from the sum of $I_3T$ and $I_5T$, with its maximum value used to scale the figure (see Supplementary Note 3.2).

relative to early mismatch elimination is due to hidden thermodynamic driving rather than to a difference between the free energy changes in the two reactions. We therefore expect pulse-like behaviour upon simultaneous addition of $I_5$ and $I_3$ to a solution of $OT$. The kinetic advantage of $I_3$ in displacing strand $O$ is matched by a corresponding increase in the rate of the reverse reaction: the position of the equilibrium, which is determined only by the free energies of the competing reactions, is relatively insensitive to the position of the mismatch. The transient excess of $I_3$ hybridised to T thus disappears as the system relaxes to equilibrium.

We challenge $OT$ with a twofold excess of $I_3$, the kinetically favoured invader, and a fourfold excess of $I_5$, which shifts the eventual equilibrium towards $I_5T$ (Fig. 6b). We indeed see an initial rapid rise in Cy3 fluorescence as $I_3$ displaces strand $O$ from the initial duplex $OT$, eliminating a position 3 mismatch. Cy3 fluorescence reaches a sharp peak before falling again as $I_3$ is itself displaced from the product duplex $I_3T$ by $I_5$ via toehold exchange[9], reaching an equilibrium that is biased towards $I_5T$. This transient behaviour shows that understanding of the kinetic properties of mismatch elimination during strand displacement can be used in the design of dynamic strand-displacement systems. Fitted second-order reaction rate constants for this system, and for a system with no mismatches, are given in Supplementary Table 1: the fits confirm that positioning the mismatch three base pairs from the $3'$ toehold increases the rate constant for invasion by $I_3$ by a factor of ~20, whereas the $I_5$ invader is accelerated by less than a factor of 2. The thermodynamic advantage of eliminating the mismatch, which is enjoyed by both the $I_5$ and $I_3$ invaders, is effectively kinetically hidden for $I_5$.

Asymmetry between the competing toeholds also affects the results: the sharpness of the transient is enhanced by an intrinsic kinetic advantage of the $I_3$ invader that is present even in the absence of mismatch elimination (see Supplementary Note 1.4, Supplementary Table 1). This asymmetry is likely to be associated with a difference in toehold stability and may be affected by the quenchers in each invader, which will influence the stability of the initial duplex[31] and may interfere with displacement initiation. Further experiments on systems with a mismatch close to the $5'$ toehold and others with inverted displacement domains (Supplementary Notes 1.4 and 3.5, Supplementary Tables 1–3, Supplementary Figs. 9–13 and 33–39) support the conclusion that the changes in rates due to mismatches are consistent with the hidden thermodynamic driving design principle proposed here, regardless of whether the mismatch works in concert with this advantage or against it.

## Discussion

Through the detailed experimental and theoretical analysis of mismatch elimination during DNA strand displacement, we have demonstrated that strand-displacement rates are substantially enhanced when a mismatch is eliminated early during displacement, by up to approximately two orders of magnitude, but the effect on reaction kinetics is minimal if the mismatch is eliminated at a later stage. The modulation of kinetics by early mismatches provides a tool for engineering simple strand-displacement networks with non-trivial dynamics. Although hybridisation reactions can also be controlled by changing toehold interaction strengths, modifying the toehold simultaneously changes both the reaction rate and the thermodynamic drive. The introduction of defect elimination allows the helpful design abstraction that rate and drive can be tuned independently. Reaction rate can be controlled by varying the position of an early mismatch without significantly affecting thermodynamic drive. More importantly, the minimal kinetic effect of later mismatches allows for hidden thermodynamic driving: the free-energy change

$\Delta G$ of a reaction can be enhanced, while leaving reaction kinetics largely unchanged. This capability is an important addition to the design tools available for the the construction of synthetic non-equilibrium systems.

A hidden thermodynamic drive makes it possible to push a reaction forwards thermodynamically without introducing longer toeholds that could significantly increase the rates of undesired reactions. Although previous work has also demonstrated the possibility of decoupling the thermodynamics and kinetics of strand displacement[9,19,32], the motifs therein do not allow for such a large increase in the overall thermodynamic drive, while maintaining the extremely low direct reaction rates that are necessary for catalytic control.

The experimentally observed behaviour, including subtle effects that lead to a non-monotonic dependence of displacement rates on mismatch position for early mismatches, is reproduced and can be explained using oxDNA simulations. These simulations confirm that the contribution to $\Delta G$ of the mismatch is largely independent of mismatch position and thus that the implementation of hidden thermodynamic driving is feasible. We have studied the mismatch repair motif extensively for one mismatch sequence. The simple biophysical understanding derived from oxDNA simulations indicates that this behaviour is robust to changes in the repair sequence used, although we would expect mismatches that are more destabilising to provide both a larger thermodynamic drive and stronger kinetic effects in certain locations. The more complete turnover observed in Fig. 5 for C–A rather than T–T mismatch elimination is consistent with the expected behaviour of a more destabilising mismatch. The generalisability of the mismatch repair motif is confirmed by our successful design of systems based on three alternative mismatch elimination sequences.

The ability to systematically engineer thermodynamic driving is important because catalytic motifs designed without it are inherently fragile. Although nearest-neighbour thermodynamic models are an excellent tool to guide the design of synthetic nucleic acid-based systems, the presence of single-stranded overhangs and fluorophore labelling can easily lead to differences in stability of a few $k_BT$ that are hard to predict[31]. These differences could substantially suppress reactions that are designed to occur with very small changes in free energy, reducing the efficacy of the networks in which they are embedded. The design freedom provided by the use of mismatch elimination in strand-displacement reactions is therefore extremely beneficial in the construction of complex networks containing catalytic motifs, in which certain reactions are deliberately kinetically frustrated until activated by hybridisation catalysis.

## Methods

**Experimental system design**. Rate assays: The mismatch type used for all single-invader mismatch-elimination experiments was a C–C mismatch, the most destabilising pairing according to the nearest-neighbour model as parameterised by SantaLucia[28]. The invader strand $I$ contains a guanine (G) at this position to form a correct Watson–Crick base pair and eliminate the C–C mismatch initially present. The $OT$ duplex toehold is a $5'$ single-stranded overhang of the $T$ strand; hence, the invading strand $I$ has its complement toehold on the $3'$ end. For comparison with the two-toehold case, we shall refer to this as $5'$ invasion.

For all mismatch positions, the mismatch motif (the mismatch and its nearest neighbours) was preserved. For mismatch positions 1–13, this was achieved by sliding the mismatch domain along by moving the base pair immediately after it to the position immediately before it (so that all sequences are identical when the mismatch domain is omitted). This method also preserves base content and most of the nearest-neighbour pairs, and causes minimal change in free energy of reactant and product duplexes according to the nearest-neighbour model[28,33]. Mismatch positions >13 occur within the domain that doubles as the reporter toehold, therefore the output sequence must remain unchanged in this region. Mismatches can only be placed at locations (15 and 17) because they are cytosines in the output strand surrounded by the correct mismatch environment. By placing a cytosine where there was a guanine in the complementary domain, the desired

mismatch motif is achieved. Specific sequences are given in Supplementary Note 2 and Supplementary Table 4.

Two-toehold competition: For the two-toehold system an identical displacement domain sequence was used for the target strand $T$ as in the single-invader experiment with the mismatch at position 3. Symmetric 4-nt sequences were added on either side of the displacement domain on $T$ as toeholds. These toeholds have a different sequence to those used in the single-toehold experiments with a higher GC content. A strand $O$, complementary to the displacement domain of $T$ except for a $C-T$ mismatch at position 18, forms an initial duplex with $T$.

A 5′ invader $I_5$ has a 3′ domain complementary to the 5′ toehold of $T$ to enable it to attack the two-toehold substrate duplex $OT$ from its 5′ toehold. The 5′ invader was labelled with 5′ Cy5 and 3′ Iowa Black RQ with 2-nt spacers between the invader strand domains and either label. Similarly, the 3′ invader $I_3$ bears the complement to the toehold domain at its 5′ end, in order to attack the two-toehold duplex from its 3′ toehold. Labels on this invader were 5′ Iowa Black FQ and 3′ Cy3 with the same spacer nucleotides before the labels. The fluorescence change observed upon displacement comes from an increase in fluorophore–quencher separation, when the invader strand is part of a duplex with the target. Sequences for all strands invovled in the two-toehold experiments are provided in Supplementary Note 2 and Supplementary Table 6.

Minimal catalytic motifs: The basic design of the system was inspired by the seesaw gate motif inspired by Winfree and Qian[34]. To introduce a systematic thermodynamic drive towards reaction completion, we modified the system to include either a hidden mismatch that is eliminated through the course of the catalytic cycle, or by extending a strand to form extra base pairs in the final complex.

In all experiments the substrate strand $D$ is identical. It contains two six-base toehold domains at either end of the strand, and a 22-base binding domain. In the default (no-mismatch) system, the output strand $B$ contains a complementary 22-base binding domain, a single toehold complementary to the 3′ end of $D$, and a single-stranded domain of 16 bases that is use to activate a fluorescent reporter. This reporter is labelled with a Cy3 moiety that is initially quenched by a complementary locking strand carrying an Iowa black quencher; fully single-stranded $B$ is able to displace the quencher, but the toehold for binding to this reporter is buried within the initial $BD$ complex. The fuel strand $C$ is identical to $B$, apart from the absence of the domain for reporter activation. The catalyst strand $A$ consists of a 22-base binding domain that is complementary to $D$, and a six-base toehold domain that is complementary to the 5′ toehold of $D$. Thus the reactions $A + BD \rightleftharpoons B + AD$ and $C + AD \rightleftharpoons A + CD$ can proceed by toehold-mediated strand exchange, with the output $B$ detected by a reporter.

In the mismatch-elimination experiments, T–T and C–A mismatches were introduced close to the centre of the 22-base binding domain. These mismatches were introduced by altering the $A$ and $B$ strands, giving variants $A_{TT}$, $B_{TT}$, $A_{CA}$ and $B_{CA}$. The same $D$, $C$ and reporter strands were used. For the experiments with additional base pairs in the final complex, strand $C_2$ was used—an extension of $C$ by two bases at its 3′ end, allowing it to form two additional base pairs with the 5′ toehold of $D$. Sequences for each strand are given in Supplementary Table 6.

**Secondary reporter systems for the rate assays**. Two different reporter systems were used to generate the rate assay results presented in this paper. The first, reporter A, had a 7-nt toehold domain as a single-stranded 5′ overhang and a 16-nt displacement domain. An additional 2 bp were present at the end of the reporter duplex opposite to the toehold. These last two base pairs are not directly invaded by the output, but will rapidly dissociate spontaneously after branch migration[9,19]. The output strand $O$ is intended to activate the reporter. Initially, the requisite toehold domain of $O$ is sequestered within the $OT$ duplex and the displacement domain exists as a single-stranded 5′ overhang.

In preliminary studies (Supplementary Fig. 2), it was discovered that reporter A is subject to leak reactions, when the mismatches to be eliminated are in the part of the $OT$ duplex that sequesters the reporter toehold domain (positions 15 and 17). In this case, the substrate duplex can displace the reporter output at a significant rate even in the absence of invader $I$, presumably due to spontaneous fraying that reveals the toehold of $O$. To avoid this effect, an alternative reporter B was designed that extended the displacement domain of reporter A by 2 bp into the toehold domain, with a compensating 2-nt 5′ extension of the toehold, and by omitting the final 2 bp of the reporter duplex such that the reporter duplex remained 18 bp in length. The result is that the 7-nt toehold for reporter B is more effectively sequestered within the $OT$ duplex, starting 2 nt from the end of the duplex. As can be seen from Supplementary Fig. 4, this change effectively eliminated the leak reactions. Unlike reporter A, reporter B cannot be used with mismatches in position 12 or 13, since the toehold of $O$ for activation of the reporter would be disturbed in these cases. Sequences, and evidence that the two reporter designs were consistent, are provided in Supplementary Notes 1 and 2, Supplementary Table 5 and Supplementary Fig. 5.

**Oligonucleotide and complex preparation**. Rate assays and two-toehold experiments: Oligonucleotides were purchased from Integrated DNA Technologies. Strands without modifications were purchased with standard desalting; no additional purification was performed. Strands with fluorophore and/or quencher modifications were HPLC purified by IDT. Strands were lyphophilised by IDT and resuspended to 100 μM in ddH$_2$O for storage. Single-stranded species were diluted

to 10 μM in fluorometry buffer FB (10 mM Tris·HCl pH 8.0 + 50 mM NaCl + 10 mM MgCl$_2$). Reporter complexes were produced by incubating reporter output and reporter target at room temperature in FB at a ratio of 6:5, determined by titration. The nominal concentration of the reporter complex is the nominal concentration of reporter target used to produce it, usually 20 μM. Output:target substrate duplexes were made by mixing components 1:1 in FB to a concentration of 10 μM and annealed by heating to 96 °C and cooling to room temperature at a rate of 6 °C min$^{-1}$.

Minimal catalytic motifs: Here, we report the methodology for experiments in which the course of the strand-exchange reaction was followed in real time, cf. Figs. 1c and 5, and Supplementary Fig. 7. Methodological differences for the systems in which reporter complexes were added after 3 and 6 h of the reaction are outlined in Supplementary Note 1.3.2. Processed data for these experiments are plotted in Supplementary Fig. 8 and unprocessed data in Supplementary Fig. 32.

Oligonucleotides were purchased from IDT at an initial concentration of 100 μM in the LabReady formulation provided by the supplier (TE buffer, pH 8.0). All strands were HPLC purified. Catalytic strands $A$, $A_{TT}$ and $A_{CA}$ were prepared at an intermediate concentration of 1 μM by diluting 1 μL of the initial stock provided by IDT in 99 μL of hybridisation buffer (TAE buffer + 1 M NaCl). A working stock of 50 nM was obtained by diluting 5 μL of this intermediate concentration in 95 μL of hybridisation buffer. The working concentrations for the fuel strands $C$ and $C_2$ were set at 10 μM by diluting 4 μL from the stock provided by IDT with 36 μL of hybridisation buffer.

$BD$, $B_{TT}D$ and $B_{CA}D$ complexes at 1 μM were prepared by adding both strands at 1.2:1 ratio (1.2 μL of $D$ and 1 μL of $B$) to 97.8 μL of hybridisation buffer. The excess of $D$ strands was used to minimise the possibility that any free $B$ strands could activate the reporter. Competition between catalyst strand $A$ and fuel strand $C$ (present in much higher concentration) ensures that free $B$ strands do not represent a significant sink for $A$. In the case of the T–T mismatch complexes, we used a ratio of 1.2:1.1 instead of 1.2:1 because titration showed that the concentration of $B_{TT}$ mismatch strand was actually relatively lower than the other $B$ strands: this ratio yielded a comparable maximal signal.

Working stocks of reporter complexes were prepared at a concentration of 2.5 μM by diluting the IDT-supplied strands in a ratio 1:1.3, with the quencher-labelled locking strand in excess to prevent invasion of $BD/B_{TT}D$ and $B_{CA}D$ complexes by free fluorophore-labelled strands. A volume of 5 μL of the fluorophore-labelled strand stock was mixed with 6.5 μL of the quencher-labelled strand stock and 188.5 μL of hybridisation buffer. The $BD$, $B_{TT}D$ and $B_{CA}D$ and the reporter–lock complexes were equilibrated by being left at room temperature for an extended period.

**Spectrofluorimetry**. Rate assays and two-toehold experiments: Fluorescence measurements were carried out in a JY Horiba Spex Fluoromax-3 spectro-fluorimeter with 4-position sample changer. For experiments involving Cy5 fluorophores, the excitation wavelength was 648 nm with a slit width of 2 nm bandpass. The emission wavelength was 664 nm with a slit width of either 1 nm, 2 nm or 5 nm bandpass in single-invader experiments and 2 nm bandpass in two-toehold experiments. For Cy3 fluorophores used in the two-toehold experiments, the excitation wavelength was 550 nm and emission wavelength was 564 nm, both with a slit width of 2 nm bandpass. Integration time for measurements was 2 s and the interval between successive measurements was 60 s. Samples were not illumi-nated between measurements to minimise photobleaching. Fluorometer tempera-ture was held at 25 °C with a water bath.

A volume of 1.92 ml samples was added to a set of four matched quartz cuvettes (Hellma) with approximate volume of 2 ml and sealed tightly with 1.5 ml Eppendorf tubes as lids to reduce photobleaching and evaporation effects by decreasing contact of sample solution with air. A volume of 2.4 μl of 20 μM reporter complex was added to 1.92 ml buffer and a fluorescence baseline recorded. After baseline collection, 2.4 μl of 10 μM output–target duplex was added and the sample mixed for ~60 s before returning to the fluorometer and collecting a second baseline. This enabled verification that there is no significant leak reaction between unreacted output–target duplex and reporter complex. After collection of a second baseline, 2.4 μl of 10 μM invader strands were added and mixed, marking $t = 0$ and the experiment start point. Raw fluorescence data are reported in the Supplementary Figs. 15–29.

For two-toehold experiments, invading strands were added to 1.92 ml FB to achieve the desired concentrations and a baseline was collected in the fluorometer. Of the four sample positions in any fluorometer run, three were used for experiments and one as a control. In the three experiment cuvettes, 2.4 μl of 10 μM $OT$ duplex was added and mixed with the solution of invaders at $t = 0$. Concurrently, the fourth cuvette with the same concentration of invaders was saturated with a 10× excess of two-toehold target strand $T$ to provide a measure of the maximum achievable fluorescence and to provide a control for global fluorescence fluctuations. Raw fluorescence data are reported in the Supplementary Figs. 33–39.

After use, cuvettes were washed 2× with 100% ethanol, then 5× with ddH$_2$O and finally 2× with ethanol again before drying with hot air and polishing the exterior with ethanol-soaked lens tissue.

Minimal catalytic motifs: Spectrofluorimetry experiments were performed in a CLARIOstar Plate Reader (BMG Labtech) using 96-well black polystyrene plates with transparent bases (SIGMA ALDRICH). The experiments used bottom optics, measuring at a fixed point in the well. Plates were shaken for 5 s before the

first measurement to ensure homogeneity in the wells. Wavelengths used for measuring the fluorescence of Cy3 were 520/30 nm for excitation and 590/20 nM for detection. The gain parameter was set to 1600 and temperature was held at 25 °C by the spectrofluorimeter's internal temperature control. Fluorescent measurements were taken each 30 s. Samples were not illuminated between the measurements and no appreciable photobleaching effect was observed.

A volume of 8 μL of $BD/B_{TT}D/B_{CA}D$ and 20 μL of reporter complex working stocks were diluted with hybridisation buffer (168 μL for the experiments without catalytic input, and 148 μL for the experiments with catalytic input) in the appropriate well. Care was taken to mix the solutions using a pipette. Fluorescence was then monitored for 20 min to check that it was consistent with the level expected for the quenched reporter. A volume of 4 μL of fuel strands $C$ or $C_2$ at the working concentration was then added and mixed as before. Fluorescence was measured for between 10 and 20 min, during which period leak reactions lead to a small increase in fluorescence, particularly for fuel strand $C_2$. Finally, 20 μL of the corresponding catalyst strands $A/A_{TT}/A_{CA}$ at the working concentration were added to the corresponding well and mixed using a 10–100 μL multichannel pipette. Experiments were left to run for 500 min, after which an excess of input (4 μL of the standard catalyst $A$ at 100 μM) was added in order to check the endpoint of the experiment and allow for the normalisation of data. Unprocessed fluorescence data are reported in the Supplementary Figs. 30 and 31.

*Data processing and fitting*. Rate assays and two-toehold experiments: Data processing and fitting was carried out using MATLAB 2014b. Fluorescence traces were processed and fitted to ordinary differential equation models based on ideal second-order kinetics; reporters were sufficiently fast (Supplementary Figs. 1 and 3) that their response was assumed to be instantaneous. To ensure that results were not sensitive to details of the processing and fitting, we used three separate approaches to estimate rate constants for displacement, as outlined in Supplementary Note 3.1. All gave essentially equivalent results, as shown in Supplementary Fig. 14. Details are provided in Supplementary Note 3.

Minimal catalytic motifs: Data were normalised separately for each experiment. We defined a baseline $F_{min}$ as the average fluorescence observed over a 10 min window prior to fuel and catalyst addition, and a maximum $F_{max}$ as the average fluorescence after addition of excess catalyst $A$, after an initial 500 s for equilibration. We plot the fractional fluorescence recovery $(F(t) - F_{min})/(F_{max} - F_{min})$ as the relative fluorescence intensity, where $F(t)$ is the measured fluorescence as a function of time.

**oxDNA methods**. oxDNA simulations were performed using the sequence-dependent version of the model presented by Sulc et al.[22] at a temperature of $T = 25$ °C. All simulations were performed on a single set of $O$, $T$ and $I$ strands in a periodic box of volume 16,686 nm³. Sequences corresponded to the experimental design.

Kinetic studies were performed using the Langevin B algorithm for rigid bodies of Davidchack et al.[35], with friction parameters set to $\gamma = 0.586$ ps⁻¹ and $\Gamma = 1.76$ ps⁻¹ and using a time step of 8.53 fs (refs. [23,24]). Forward Flux Sampling[36,37] was used to enhance sampling of the displacement process, with individual dynamical trajectories used to infer the conditional probabilities reported in Fig. 3b. Details of the FFS, and the statistics of sampling runs, are reported in Supplementary Note 4 and Supplementary Tables 8–12. Relative fluxes for different systems give a good estimate of the relative rate constants obtained in the dilute second-order limit, as discussed in refs. [19,23,38].

Thermodynamic sampling, used to obtain the free-energy profile and overall free energies of reactions, was performed using the Virtual-Move Monte Carlo method of Whitelam and Geissler[39,40] (specifically, the variant in the appendix of ref. [40]). Seed moves were the rotation of a nucleotide about its backbone site (angle drawn from a Gaussian distribution with standard deviation 0.12 radians) and translation of a nucleotide (distance drawn from a Gaussian distribution with standard deviation 0.102 nm). Umbrella Sampling[41] was employed to enhance sampling efficiency. Details of the umbrella sampling procedure are provided in Supplementary Note 4 and Supplementary Tables 13–22.

**Reporting summary**. Further information on research design is available in the Nature Research Reporting Summary linked to this article.

## Data availability

The data generated during all the experiments and simulations reported and analysed in this manuscript are available in the Zenodo repository, https://doi.org/10.5281/zenodo.3731499. Any other relevant data are available from the authors upon reasonable request.

## Code availability

The code used to fit experimental data to ODE models is available in the Zenodo repository, https://doi.org/10.5281/zenodo.3731499. Instructions on how to reproduce the plots and fits from the raw data are included. The oxDNA simulation package is available for download as standalone software from https://dna.physics.ox.ac.uk/, and as the LAMMPS USER-CGDNA package from http://lammps.sandia.gov (ref. [42]). The former package can be used to implement the VMMC-based umbrella sampling simulations reported here, with the simulation parameters specified in Supplementary

Note 4. The Langevin-based FFS can be performed with the LAMMPS implementation, using the simulation parameters specified in Supplementary Note 4.

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

## Acknowledgements
N.E.C.H. was supported by Engineering and Physical Sciences Research Council grant EP/F500394/1, A.J.T. by a Royal Society Wolfson Research Merit Award and T.E.O. by a Royal Society University Research Fellowship.

## Author contributions
N.E.C.H., T.E.O., J.B. and A.J.T. conceived the project, following initial computational investigations by A.G. under the supervision of A.A.L. and T.E.O. N.E.C.H., I.M.R., T.E.O., J.B. and A.J.T. planned the experiments and simulations. N.E.C.H. and I.M.R. performed the experiments and analysis; T.E.O. performed the simulations and analysis. N.E.C.H., I.M.R., T.E.O., J.B. and A.J.T. wrote the paper.

## Competing interests
The authors declare no competing interests.
