## [Peer Review File · Nature Communications]

Reviewers' Comments:

Reviewer #1:

Remarks to the Author:

The authors demonstrate in this manuscript the establishment of a "hidden" thermodynamic drive in toehold-mediated strand displacement reactions by incorporating a mismatch into the incumbent-target duplex. This favors the final displaced state. Additionally, this allows to modulate the reaction kinetics. While mismatches at the toehold-distal end barely affect the displacement rate, a strong acceleration is observed for mismatches near the toehold. Using coarse-grained molecular dynamics simulations of the reaction, the authors can reproduce the experimental findings in silico and furthermore provide a qualitative model for the experimental observation. Additionally, the authors show that they can apply their system as a hidden thermodynamic drive in catalytically triggered strand displacement reactions as well as thermodynamically neutral control to generate a pulse pattern in a competition between two exchange reactions. Overall, the presented results will be very useful for the field of DNA-based reaction networks.

Detailed comments:

- 1) The authors emphasize very much that their system represents a "hidden" thermodynamic drive that does not change the reaction kinetics. One of their examples, the pulse generation, does however not use the drive but rather the modulation of the reaction kinetics dependent on the mismatch position. I suggest to highlight this two-fold usage of their mechanism, e.g. in the discussion.
- 2) The authors state in section 1.1 that "The reaction rate constant rises as the mismatch is moved away from the toehold up to position 3" and call this "striking". I suggest to tone down somewhat the statement, since this non-monotonic behavior is provided by a single data point.
- 3) The data plots in Figs 2b and 3b are somewhat hard to read. Since the figures contain already color, I suggest the usage of colors also in the data plots that allow better to differentiate between exp. and simulation data or between the mismatch and the mismatch-free system.
- 4) The newly added Section 1.3 interrupts somewhat the red-line of the manuscript and is also more a detailed "expert-discussion". It certainly has its merits but placing it in the SI would also suffice.
- 5) The C2 system mentioned in Figure 1 should be shown as well (at least in SI) to make the manuscript more accessible for non-experts.

Reviewer #2:

Remarks to the Author:

Overall, I think this is an interesting study that introduces a potentially useful tool for DNA strand displacement. Beyond a specific design tool, I also think some of the insights and tabulated rate constants will be useful for reference in situations where mismatch elimination might arise in a number of applications. Further, the authors' claims are well supported and the work is very solid scientifically. That being said I am not entirely sure this study meets the broad interest and novelty criteria for publication in *Nature Communications* and might be potentially be more appropriate for a journal like *Nucleic Acids Research* – especially given the first four figures are devoted to a very detailed and thorough experimental and theoretical analysis of the influence of mismatch position on kinetics. Additionally, the authors have previously presented the ability to design transient pulse-like

dynamics through mismatch creation – although there are subtle differences between the previously published mechanism and the mechanism presented in this paper the final result between this study and that previous study are very similar.

I do think the authors did a thorough job of addressing the previous reviewers' points. I do have two comments below where I think the authors responses could be improved further:

- Reviewer 2 in Comment 1c brings up the interesting point of having bulkier bases for the mismatches (AA or GG) and how this might influence the rate of displacement. The authors successfully demonstrate multiple mismatch pairs but most of them are between two pyrimidine bases (with the exception of CA -> CG). It might be insightful for the authors to comment on whether the size of the bases in the mismatch influence the stability of the surrounding duplex and thus influence the rate of displacement. Also, the authors don't really discuss how different sequences can influence kinetics (see Comment 17 below) – I agree explicitly studying this is outside of the scope of the current manuscript but in the discussion I think it is important to note that different mismatch sequences can behave differently.

- For Minor Point 3 of Reviewer 2, I agree with the reviewer that mismatch repair could be confused with enzymatic DNA repair mechanisms – however the authors still have the term mismatch repair throughout the main text and a few times in the SI despite responding that they changed this term to "mismatch elimination"

I do have a number of additional comments, many of which I believe would improve the clarity of the figures and narrative.

Minor comments and suggestions

- 1) Last sentence of the second paragraph in the introduction has a typo: thexerfore -> therefore
- 2) In Figure 1c the reporting complex is missing half arrows that show strand directionality i.e. the 3' ends of the strands are not designated as in Figure 1a and Figure 1b.
- 3) In Figure 1c I think showing C2 above or below C would make the figure much clearer, even with the description in the figure caption.
- 4) Why are there two B molecules shown in Figure 1b going to the reporter? I think this is supposed to depict the first B comes from A reacting with BD and the second is from C reacting with BD (the leak reaction). I think it would clarify the figure if this were explicitly written somewhere on the figure.
- 5) In Figure 1a the strand A is the invading strand and then in Figure 1b A is the Catalyst and C is analogous to the invading strand. As a suggestion, I think might be clearer to switch the names of the molecules in Figure 2b – i.e. make the Catalyst C and the invading strand A.
- 6) What is the concentration of the Reporter for the experiments in Figure 1? 40 nM to match the total amount of B that could be produced? This is important to understand how close the actual reaction is to completion. I couldn't find the concentration of the reporter used – it's possible I missed it but adding it to the figure caption would be good.
- 7) In the Figure 2 caption, the authors state the k_o is 2.5×10^3 but in the actual text they state it is 2.4×10^3 . Unify these two numbers.
- 8) For Section 1.3 I think the title of "Generalization to alternative toehold lengths" would be clearer. I initially interpreted alternative toeholds to potentially mean different sequences.
- 9) In Section 1.3 the authors consistently use the term "regime" except in the last paragraph they use the term "case (iii)". For consistency I suggest updating to "regime (iii)".
- 10) In Figure 5 it would be nice to label the exact position of the mismatch for easy comparison to the previous results. Also the captions in b and c would be clearer as "with catalyst (+A)" and "without

catalyst (-A)" or something that incorporates the strand name from a. Also I suggest using the nomenclature C2 rather than +2 bp in Figure 5 for consistency with Figure 1 and the text.

More in-depth comments and questions

11) Mismatches at the beginning of the duplex (positions 2 and 3) are basically like increasing the toehold length are they not?. A mismatch at position 2 is going to be like a 6-base toehold and a mismatch at position 3 is going to be like a 7-base toehold as the 1-2 bases at the 3' end of the O strand are not likely to be stably hybridized for these mismatch positions. Would this explain the non-monotonicity the authors comment on in Section 1.1? A short discussion of this might be insightful.

12) For the results in Figure 3 it looks like the $p(x \text{ reached} | \text{toehold})$ is only increased until the mismatch position reaches 7-bases and then this probability becomes the same as without a mismatch yet the authors don't highlight this point in their discussion in the text and make the general claim that $p(x \text{ reached} | \text{toehold})$ increases with increasing mismatch positions. After reading further, I see the authors later clarify that from position 7 on the rate of reaching the mismatch is not increased – I suggest for clarity that they make this distinction at the point in the text where they state "We see that the mismatch at x enhances both the probability that base pair x is reached and the probability of overall success given that x has been reached."

13) When the Catalyst has a mismatch you also have the rate slowing effect of branch migration through the mismatch domain so this is confounding to the simple results in the previous analysis. I imagine the location of this mismatch has consequences beyond just the analysis presented previously in this paper. For example, if the mismatch is at either end then the leak would be increased or if at early positions in BD then the A invasion would be much slower than having the mismatch in the middle or later...I think a discussion of this would be beneficial in the text.

14) For Figure S8 the caption says that 11.1 μM A, 44.4 μM BD, and 55.7 μM of C were used to initiate the experiment. These concentrations are 3 orders of magnitude higher than the concentrations used in Figure 5 but the methods in the SI state that the experiments were run similarly to Figure 5. Is there a typo here or these higher concentrations were used? One thing that doesn't add up is the CA mismatch data in Figure 5b indicates the reaction takes over 24 hours to complete yet in Figure S8 the authors claim that incubating the reaction for 3 hours before adding the reporter is enough for the reaction to essentially reach full completion – how can this be? Perhaps this is why the authors used the higher concentrations for the experiments in Figure S8? I think clarifying what is going on here is key for the reader to make appropriate conclusions about the results – if higher concentrations were used to speed up the reactions then this should be stated so the reader knows the times in Figure 5 and Figure S8 are not comparable. If these higher concentrations were used, I also wonder how having 1000x more of each species influences the leak reaction?

15) In the last paragraph of the of Section 1.5, the authors claim that the difference in fluorophores used on I3 and I5 could give rise to different stabilities in initial toehold binding, however, the quenchers of each of the strands are what are actually present to provide the stability for binding of I3 or I5 to the OT complex– their argument is still valid but they should change fluorophores -> quenchers I believe (based on the diagram in Figure 6).

16) Can the results in Figure 6 be reproduced without a mismatch but using different toehold lengths, say a k_5 of 4 bases and k_3 of 6 bases where I5 invades all of the way to the 5' end of the O domain and I3 invades only to 2 bases from the 3' end of the O domain? Both invaded structures are the same thermodynamically but I3 is kinetically favored initially? A comment on if this is possible and a comparison of the strengths and weaknesses of this approach vs the mismatch approach would be illuminating.

17) The authors demonstrate that they can use a variety of different mismatches for successful results – however it does appear that the sequence influences the kinetics, CA -> CG was faster than TT -> TA. I think it would be beneficial to comment on this as a potential design parameter to be considered. Do the authors think this is due to CG being more stable than TA during the repair or?

18) In Table S1 is there a typo for CG -> GG introduction k_5 rate? How can it be 7 orders of

magnitude lower than k_3 in this case? That also isn't consistent with the changes in rates due to mismatch introduction presented in Table S2.

REVIEWER 1

We thank the reviewer, who noted that “the presented results will be very useful for the field of DNA-based reaction networks”, for their supportive comments. With regard to the

COMMENT 1: *“The authors emphasize very much that their system represents a ”hidden” thermodynamic drive that does not change the reaction kinetics. One of their examples, the pulse generation, does however not use the drive but rather the modulation of the reaction kinetics dependent on the mismatch position. I suggest to highlight this two-fold usage of their mechanism, e.g. in the discussion”*

REPLY: This is a good suggestion.

ACTION: We have adjusted the first paragraph of the conclusion as outlined below.

Through detailed experimental and theoretical analysis of mismatch elimination during DNA strand displacement, we have demonstrated that strand displacement rates are substantially enhanced when a mismatch is eliminated early during displacement, by up to approximately two orders of magnitude, but the effect on reaction kinetics is minimal if the mismatch is eliminated at a later stage. **The modulation of kinetics by early mismatches provides a tool for engineering simple strand displacement networks with non-trivial dynamics. Although hybridization reactions can also be controlled by changing toehold interaction strengths, modifying the toehold simultaneously changes both the reaction rate and the thermodynamic drive. The introduction of defect elimination allows the helpful design abstraction that rate and drive can be tuned independently. Reaction rate can be controlled through the position of an early mismatch without significantly affecting thermodynamic drive. More importantly, the minimal kinetic effect of later mismatches allows for hidden thermodynamic driving: the free-energy change ΔG of a reaction can be enhanced whilst leaving reaction kinetics largely unchanged. This capability is an important addition to the design tools available for the the construction of synthetic non-equilibrium systems.**

COMMENT 2: *“The authors state in section 1.1 that ”The reaction rate constant rises as the mismatch is moved away from the toehold up to position 3” and call this ”striking”. I suggest to tone down somewhat the statement, since this non-monotonic behavior is provided by a single data point.”*

ACTION: We have removed the adjective “striking” in the comparison to previous work, now only noting that the two differ.

COMMENT 3: *The data plots in Figs 2b and 3b are somewhat hard to read. Since the figures contain already color, I suggest the usage of colors also in the data plots that allow better to differentiate between exp. and simulation data or between the mismatch and the mismatch-free system.*

REPLY: We agree with the reviewer’s comment.

ACTION: We have edited Fig. 2b so that simulation results are plotted with solid symbols. In Fig 3b, we have used blue, green and orange to distinguish curves.

COMMENT 4: *“The newly added Section 1.3 interrupts somewhat the red-line of the manuscript and is also more a detailed ”expert-discussion”. It certainly has its merits but placing it in the SI would also suffice.”*

REPLY: On balance, we believe that the discussion (which addresses comments of previous reviewers) is best in the main text, and so we propose to leave it there.

COMMENT 5: *The C2 system mentioned in Figure 1 should be shown as well (at least in SI) to make the manuscript more accessible for non-experts.*

REPLY: We agree with the reviewer.

ACTION: Both C and C2 are now shown in the Fig. 1b.

REVIEWER 2

We thank the reviewer for their careful assessment of our manuscript, extensive advice and positive comments about our “interesting” and “potentially useful” study that is “very solid scientifically”. In terms of novelty, we note that the primary focus of the manuscript is on introducing and demonstrating a realisation of hidden thermodynamic driving, which is not a feature of the previous study to which the reviewer refers. It is this aspect of our work that makes it appropriate for Nature Communications. The details of the implementation of hidden thermodynamic driving through nucleic acids are important in setting up our approach as a generally applicable tool; indeed, previous reviewers wanted *more* detail of this kind for publication in Nature Chemistry!

COMMENT A: *“Reviewer 2 in Comment 1c brings up the interesting point of having bulkier bases for the mismatches (AA or GG) and how this might influence the rate of displacement. The authors successfully demonstrate multiple mismatch pairs but most of them are between two pyrimidine bases (with the exception of CA -> CG). It might be insightful for the authors to comment on whether the size of the bases in the mismatch influence the stability of the surrounding duplex and thus influence the rate of displacement. Also, the authors don't really discuss how different sequences can influence kinetics (see Comment 17 below) I agree explicitly studying this is outside of the scope of the current manuscript but in the discussion I think it is important to note that different mismatch sequences can behave differently.”*

REPLY: A number of factors come into play in determining the destabilizing effect of a mismatch – not least the stability of the intact base pair to which it is compared. We do not believe that there is overwhelming evidence to support the idea that bulkier bases are systematically more destabilizing - indeed, the C-C mismatch used here is the most destabilizing according to the widely-used SantaLucia model. We therefore don't want to spend too much time hypothesising about which mismatches are more destabilizing. However, it is true that mismatches that are less destabilizing will tend to produce a smaller free-energetic drive, which is perhaps consistent with the differences between C-A and T-T in Fig. 5b.

ACTION: The second paragraph of the Discussion has been augmented to include the following:

We have studied the mismatch repair motif extensively for one mismatch sequence. The simple biophysical understanding derived from oxDNA simulations indicates that this behaviour is robust to changes in the repair sequence used, although we would expect mismatches that are more destabilizing to provide both a larger thermodynamic drive and stronger kinetic effects in certain locations. The more complete turnover observed in Fig. 5 for C-A rather than T-T mismatch elimination is consistent with the expected behaviour of a more destabilizing mismatch. The generalizability of the mismatch repair motif is confirmed by our successful design of systems based on three alternative mismatch elimination sequences.

COMMENT B: *“For Minor Point 3 of Reviewer 2, I agree with the reviewer that mismatch repair could be confused with enzymatic DNA repair mechanisms however the authors still have the term mismatch repair throughout the main text and a few times in the SI despite responding that they changed this term to “mismatch elimination””*

ACTION: We have searched again and updated all the examples of “repair” we could find.

COMMENT 1: *“Last sentence of the second paragraph in the introduction has a typo: therefore -> therefore.”*

ACTION: Typo fixed.

COMMENT 2: *“In Figure 1c the reporting complex is missing half arrows that show strand directionality*

i.e. the 3' ends of the strands are not designated as in Figure 1a and Figure 1b."

ACTION: Figure fixed.

COMMENT 3: *"In Figure 1c I think showing C2 above or below C would make the figure much clearer, even with the description in the figure caption. "*

REPLY: We agree with the reviewer.

ACTION: Done.

COMMENT 4: *" Why are there two B molecules shown in Figure 1b going to the reporter? I think this is supposed to depict the first B comes from A reacting with BD and the second is from C reacting with BD (the leak reaction). I think it would clarify the figure if this were explicitly written somewhere on the figure. "*

REPLY: It wasn't our intention to indicate the leak in this manner.

ACTION: We have clarified the figure by only boxing one of the output B strands.

COMMENT 5: *" In Figure 1a the strand A is the invading strand and then in Figure 1b A is the Catalyst and C is analogous to the invading strand. As a suggestion, I think might be clearer to switch the names of the molecules in Figure 2b i.e. make the Catalyst C and the invading strand A. "*

REPLY: We respectfully disagree with this suggestion. Firstly, catalyst and invading strands are not mutually exclusive sets; our catalysis mechanism involves two separate strands (one of them the catalyst) acting as invaders. Secondly, there is no catalysis shown in Fig. 1a of Fig. 2b, so we think it would be misleading to label the figures as such.

COMMENT 6: *"What is the concentration of the Reporter for the experiments in Figure 1? 40 nM to match the total amount of B that could be produced? This is important to understand how close the actual reaction is to completion. I couldn't find the concentration of the reporter used its possible I missed it but adding it to the figure caption would be good. "*

REPLY: The concentration of the reporters was 250 nM (large excess). This information was buried in the methods.

ACTION: We have added the concentration of the reporter to the caption.

COMMENT 7: *" In the Figure 2 caption, the authors state the k_0 is 2.5×10^3 but in the actual text they state it is 2.4×10^3 . Unify these two numbers."*

ACTION: Unified.

COMMENT 8: *"For Section 1.3 I think the title of "Generalization to alternative toehold lengths" would be clearer. I initially interpreted alternative toeholds to potentially mean different sequences."*

ACTION: Done.

COMMENT 9: *" In Section 1.3 the authors consistently use the term regime except in the last paragraph they use the term "case (iii)". For consistency I suggest updating to "regime (iii)"."*

ACTION: Done.

COMMENT 10: *“In Figure 5 it would be nice to label the exact position of the mismatch for easy comparison to the previous results. Also the captions in b and c would be clearer as “with catalyst (+A)” and “without catalyst (-A)” or something that incorporates the strand name from a. Also I suggest using the nomenclature C2 rather than +2 bp in Figure 5 for consistency with Figure 1 and the text.”*

REPLY: We concur with these suggestions, but note that the mismatch is not in exactly the same location for the CA and TT cases; we prefer then to simply state its location in the caption.

ACTION: We have updated the titles of the graphs to include “(+A)” and “(-A)”, and changed the legend to C2. We have also added “**These mismatches are positioned 12 and 11 base pairs, respectively, from the end of the AD duplexes.**” to the caption.

COMMENT 11: *“Mismatches at the beginning of the duplex (positions 2 and 3) are basically like increasing the toehold length are they not?. A mismatch at position 2 is going to be like a 6-base toehold and a mismatch at position 3 is going to be like a 7-base toehold as the 1-2 bases at the 3 end of the O strand are not likely to be stably hybridized for these mismatch positions. Would this explain the non-monotonicity the authors comment on in Section 1.1? A short discussion of this might be insightful.”*

REPLY: We are reluctant to push this analogy too far. Indeed, mismatches at positions 2 and 3 cause fraying, but not 100% of the time. In our simulations, in fact, the mismatch at position 3 does not cause an overwhelming tendency to fray. The point is that, even so, the final base pairs in the incumbent are more weakly attached than the base pairs of the toehold of the invader. So a toehold-bound invader is extremely likely to succeed in displacing these base pairs and enclosing the mismatch, at which point displacement is likely to succeed. Given these subtleties, we prefer our approach based on the detailed analysis of the oxDNA results, rather than simpler analogies.

COMMENT 12: *“For the results in Figure 3 it looks like the $p(x \text{ reached—toehold})$ is only increased until the mismatch position reaches 7-bases and then this probability becomes the same as without a mismatch yet the authors dont highlight this point in their discussion in the text and make the general claim that $p(x \text{ reached—toehold})$ increases with increasing mismatch positions. After reading further, I see the authors later clarify that from position 7 on the rate of reaching the mismatch is not increased I suggest for clarity that they make this distinction at the point in the text where they state “We see that the mismatch at x enhances both the probability that base pair x is reached and the probability of overall success given that x has been reached.””*

REPLY: We see that this text could be misleading, as the reviewer notes.

ACTION: We have changed the text to “We see that, **depending on the value of x , a mismatch at x can enhance** both the probability that base pair x is reached and the probability of successful displacement given that x has been reached.”

COMMENT 13: *“When the catalyst has a mismatch you also have the rate slowing effect of branch migration through the mismatch domain so this is confounding to the simple results in the previous analysis. I imagine the location of this mismatch has consequences beyond just the analysis presented previously in this paper. For example, if the mismatch is at either end then the leak would be increased or if at early positions in BD then the A invasion would be much slower than having the mismatch in the middle or later I think a discussion of this would be beneficial in the text. ”*

REPLY: The catalyst doesn’t introduce a new mismatch, it simply retains one that was present in the BD complex. When a mismatch is replaced by a mismatch then the possibility of a metastable intermediate state, with the mismatch unpaired and the invader and incumbent strands bound either side of it, exists. We have not found evidence that this putative metastable state is a large barrier to completing displacement; indeed, deep in the second order limit, one would expect its effects are pretty marginal. Locating the mismatch near the ends of either the BD or AD duplexes would indeed encourage leak reactions, but we believe that this possibility is simply a manifestation of the effects discussed earlier in the paper.

ACTION: We have updated the discussion of the catalytic system in the following way:

“Both mismatch elimination systems show substantially stronger rate enhancement in response to the introduction of the catalyst than does the default system with perfectly matched sequences. We see that hybridization catalysis is effective in this system even though the intermediate formed between strand *D* and catalyst *A* retains the initial mismatch, at least in the dilute limit with centrally- placed mismatches. In fact, mismatch-elimination systems are substantially faster overall, faster even than mismatch-free system with the longer strand *C*₂ that forms more base pairs with *D* (cf. Fig. 1). Crucially, however, the leak reaction in the absence of the catalyst remains weak, much weaker than when the longer strand *C*₂ is used (Fig. 5(c)), confirming that the thermodynamic drive is indeed successfully hidden. A second replica of the same experiments, with similar results, is reported in Supplementary Fig. S7.”

COMMENT 14: “For Figure S8 the caption says that 11.1 μ M *A*, 44.4 μ M *BD*, and 55.7 μ M of *C* were used to initiate the experiment. These concentrations are 3 orders of magnitude higher than the concentrations used in Figure 5 but the methods in the SI state that the experiments were run similarly to Figure 5. Is there a typo here or these higher concentrations were used? One thing that doesnt add up is the *CA* mismatch data in Figure 5b indicates the reaction takes over 24 hours to complete yet in Figure S8 the authors claim that incubating the reaction for 3 hours before adding the reporter is enough for the reaction to essentially reach full completion how can this be? Perhaps this is why the authors used the higher concentrations for the experiments in Figure S8? I think clarifying what is going on here is key for the reader to make appropriate conclusions about the results if higher concentrations were used to speed up the reactions then this should be stated so the reader knows the times in Figure 5 and Figure S8 are not comparable. If these higher concentrations were used, I also wonder how having 1000x more of each species influences the leak reaction?”

REPLY: This is a typo; the concentrations should be in nM. With regard to the timescales of reaction, there is another typo, and we thank the reviewer for spotting it! The axes should be labelled with $\times 10^3$ s, not $\times 10^4$ s.

ACTION: Typos have been fixed in Figs 2, 5, S7-S8, and S30-S32.

COMMENT 15: “In the last paragraph of the of Section 1.5, the authors claim that the difference in fluorophores used on *I3* and *I5* could give rise to different stabilities in initial toehold binding, however, the quenchers of each of the strands are what are actually present to provide the stability for binding of *I3* or *I5* to the *OT* complex their argument is still valid but they should change fluorophores - \dot{z} quenchers I believe (based on the diagram in Figure 6).”

REPLY: We thank the reviewer for pointing this out.

ACTION: Fixed.

COMMENT 16: “Can the results in Figure 6 be reproduced without a mismatch but using different toehold lengths, say a *k5* of 4 bases and *k3* of 6 bases where *I5* invades all of the way to the 5 end of the *O* domain and *I3* invades only to 2 bases from the 3 end of the *O* domain? Both invaded structures are the same thermodynamically but *I3* is kinetically favored initially? A comment on if this is possible and a comparison of the strengths and weaknesses of this approach vs the mismatch approach would be illuminating.”

REPLY: The proposed design would actually be more similar to the system reported in Ref. 19. To see why, note that the *k3* invader would effectively create a 6-bp toehold for *k5*, meaning that the displacement of *k3* by *k5* would be much faster than the invasion of the initial complex by *k5*, comparable to the rate of the initial *k3* invasion. The result is a much smoother non-monotonic feature, rather than the sharp spike. A variant could be made where *k5* binds to two bases from the toehold of *k3* instead, which would be more like the system discussed here. The relative advantages of achieving similar effects by modulating toeholds and mismatch location are discussed in Ref. 19, and are not the primary focus of this work; a key point is that designs of the kind proposed by the reviewer disrupt the convenient abstraction of a clear division

between toehold and branch migration domains.

ACTION: We have added a short note on the first paragraph of the conclusion: “The modulation of kinetics by early mismatches provides a tool for engineering simple strand displacement networks with non-trivial dynamics. Similar effects could be achieved by using toeholds and displacement domains of variable length in an experiment, at the expense of weakening the helpful abstraction that toehold and displacement domains can be treated separately.”

COMMENT 17: “The authors demonstrate that they can use a variety of different mismatches for successful results however it does appear that the sequence influences the kinetics, CA to CG was faster than TT to TA. I think it would be beneficial to comment on this as a potential design parameter to be considered. Do the authors think this is due to CG being more stable than TA during the repair or?”

REPLY: As the reviewer noted, there are some subtleties with the kinetics of the catalytic system involving mismatch elimination, related to the initial displacement in which the mismatch is retained. However, it does not surprise us that C-A to C-G mismatch repair gives a stronger thermodynamic push, given both the extra stability of the C-G base pair and the weaker interaction of the T-T mismatch.

ACTION: We have added the following text to the conclusion:

“...the simple biophysical understanding derived from oxDNA simulations indicates that this behaviour is robust to changes in mismatch sequence used, although we would expect mismatches that are more unstable compared to the perfectly-matched counterpart to provide a larger thermodynamic drive, and stronger kinetic effects in certain locations. This generalizability is confirmed by our successful design of systems based on three alternative mismatch elimination sequences, and the more complete turnover observed in Fig. 5 for C-A rather than T-T mismatch elimination is consistent with the expected behaviour of a more destabilizing mismatch.”

COMMENT 18: “In Table S1 is there a typo for CG to GG introduction k_5 rate? How can it be 7 orders of magnitude lower than k_3 in this case? That also isnt consistent with the changes in rates due to mismatch introduction presented in Table S2.”

REPLY: This parameter is effectively unmeasurably small; its value means no more than $k_5 \ll k_3, k_{35}, k_{53}$. I_5 initially reacts very slowly, but because $k_3 \gg k_5$, a pool of I_3T complexes quickly builds up. I_5 reacts quickly with this pool, masking any signal from the direct displacement of T by I_5 .

ACTION: We have added a note to the caption indicating that the precise value of this rate constant should not be trusted; the only salient fact is that it is small.